# Avidity of IgG to SARS-CoV-2 RBD as a Prognostic Factor for the Severity of COVID-19 Reinfection

**DOI:** 10.3390/v14030617

**Published:** 2022-03-16

**Authors:** Victor Manuylov, Olga Burgasova, Olga Borisova, Svetlana Smetanina, Daria Vasina, Igor Grigoriev, Alexandra Kudryashova, Maria Semashko, Bogdan Cherepovich, Olga Kharchenko, Denis Kleymenov, Elena Mazunina, Artem Tkachuk, Vladimir Gushchin

**Affiliations:** 1Gamaleya National Research Center for Epidemiology and Microbiology, 123098 Moscow, Russia; olgaburgasova@mail.ru (O.B.); d.v.vasina@gmail.com (D.V.); iggrigoriev.ltb@gmail.com (I.G.); maria.a.semashko@gmail.com (M.S.); 10000let@rambler.ru (D.K.); lenok27microb@gmail.com (E.M.); artem.p.tkachuk@gmail.com (A.T.); wowaniada@gmail.com (V.G.); 2MedipalTech LLC, 141981 Dubna, Russia; 3Medical Institute, Peoples Friendship University of Russia (RUDN University), 117198 Moscow, Russia; 4Mechnikov Research Institute for Vaccines and Sera, 105064 Moscow, Russia; olga.v.borisova@gmail.com (O.B.); 2238250@rambler.ru (A.K.); cherepovichb@mail.ru (B.C.); bio139@yandex.ru (O.K.); 5Infectious Disease Clinical Hospital № 1, 125367 Moscow, Russia; ikb1@zdrav.mos.ru

**Keywords:** SARS-CoV-2, COVID-19 reinfection, IgG avidity

## Abstract

The avidity index (AI) of IgG to the RBD of SARS-CoV-2 was determined for 71 patients with a mild (outpatient) course of COVID-19, including 39 primarily and 36 secondarily reinfected, and 92 patients with a severe (hospital) course of COVID-19, including 82 primarily and 10 secondarily infected. The AI was shown to correlate with the severity of repeated disease. In the group of outpatients with a mild course, the reinfected patients had significantly higher median AIs than those with primary infections (82.3% vs. 37.1%, *p* < 0.0001). At the same time, in patients with a severe course of COVID-19, reinfected patients still had low-avidity antibodies (median AI of 28.4% vs. 25% in the primarily infected, difference not significant, *p* > 0.05). This suggests that the presence of low-avidity IgG to RBD during reinfection is a negative prognostic factor, in which a patient’s risk of developing COVID-19 in a severe form is significantly increased. Thus, patients with IgG of low avidity (AI ≤ 40%) had an 89 ± 20.5% chance of a severe course of recurrent COVID-19, whereas the detection of high-avidity antibodies (AI ≥ 50%) gave a probability of 94 ± 7.9% for a mild course of recurrent disease (*p* < 0.05).

## 1. Introduction

According to published data, cases of COVID-19 reinfection are not uncommon. In particular, the surveillance of healthcare workers in the U.K. [1,2] and Denmark [3], carried out in 2020 (i.e., before vaccinations began), showed that from 0.15% to 0.87% of patients who were seropositive after the initial COVID-19 infection were reinfected within about 6 months of follow-up. Similar data were obtained for a large cohort of patients (not at risk) in the USA [4]; approximately 0.8% had a new positive PCR test for SARS-CoV-2 within 270 days (but not before 90 days) after initial recovery. Furthermore, patients with reinfection were more likely to need hospital admission than patients with primary COVID-19 [4]. Among healthcare workers in Chicago, USA [5], up to 2.5% presented a probable reinfection within 6 months of follow-up.

The risk factors for reinfection and, more importantly, its outcome (i.e., the prognosis factors for the severity of recurrent disease) are still poorly understood [4,6,7]. It is reasonable to assume that the main risk factor for repeated infection is an insufficiently protective immunity. Indeed, we already know that acquired immunity to SARS-CoV-2 and other beta-coronaviruses is not lifelong; it retains its protective properties, according to various data, for 6–30 months [8,9,10]. However, the “protective” humoral immunity usually means some quantitively measured level of class G antibodies (IgG) or, more commonly, a virus-neutralizing titer of antibodies to Spike/RBD (receptor-binding domain) epitopes of SARS-CoV-2 [11,12], which are not suitable for retrospective studies of patients who are already reinfected. In COVID-19, the IgG titer increases rapidly in the very early stages of the disease (5–7 days after infection [13]). When a symptomatic patient comes to the investigator, it is no longer possible to determine how many antibodies he or she had before the reinfection. 

However, there is at least one dynamic parameter of immunity that remains the same during the reinfection process and therefore may be used in retrospective studies. This is antibody avidity, a measure of the cooperative affinity (binding strength) of IgG and the antigen [14,15]. It increases over time, following the maturation of B-lymphocytes [13,16]—the process which takes three to four months after immunization [10,13,17,18,19,20,21]. If re-exposed to the antigen, B-lymphocytes already produce high-avidity antibodies, making it possible to distinguish between primary and secondary infection, how it is used for many viral infections, such as rubella [14,22], cytomegalovirus [23], Dengue [24], or Zika infection [25,26]. 

For COVID-19, the IgG avidity is interesting not so much as an indicator of primary or secondary disease, but as a possible prognostic marker of the course and outcome of repeated infection. IgG avidity (more specifically, the affinity of particular immunoglobulins to certain epitopes of the RBD and S1-domain [27,28]) plays a role in their virus neutralization ability [29]. Some antibodies can bind the RBD in a way that blocks its interaction with human cell receptor ACE2 and thus exert the neutralizing activity against the virus [30,31]. Yet, the RBD–ACE2 complex itself has an extremely high thermodynamic binding constant [32]. Thus, the neutralizing antibody must have at least higher affinity to RBD than the ACE2 protein to compete effectively for binding of the virus antigen. Based on this, a number of authors [33,34] conclude that only IgG with high avidity (or, more precisely, the high-affinity fraction of the pool of all IgG to the RBD, the proportion of which means the avidity index [27]) are significant for the virus-neutralizing effectiveness of serum. Direct correlation between avidity index and the titer of neutralizing antibodies was shown in [35].

It is known that in some immunized patients, IgG do not acquire high avidity [34,36] even after the time that is needed for B-lymphocyte maturation (3–4 months) [18,19,20]. It can be assumed that the adaptive immune system of these individuals has failed to mature B-lymphocytes; the proportion of high-affinity (and, therefore, virus-neutralizing) IgG in their serum is low, and they are at risk of reinfection and/or its negative outcome.

In this paper, we present data from a retrospective study of reinfected SARS-CoV-2 patients (based on their medical history) and their antibody avidity to support the assumption that reinfection with COVID-19 in the presence of insufficient IgG avidity correlates with a more severe course of the disease.

## 2. Materials and Methods

*IgG to the SARS-CoV-2 RBD* in blood serum samples was determined using the “SARS-CoV-2-RBD-Gamalya” ELISA reagent kit produced by the Gamaleya Research Center, Russia (registered for medical use in Russia, certificate No. RZN 2020/10393, dated 11 September 2020). The result for IgG presence in this kit is numbered in off-system PC units (positivity coefficient) proportional to the number of antibodies in the sample. A sample with PC ≥ 1.1 is considered positive.

*The avidity index of IgG to SARS-CoV-2 RBD* in blood serum samples was determined using the “SARS-CoV-2-IgG plus” reagent kit manufactured by MedipalTech, Russia (certificate for medical use No. RZN 2021/14424, dated 27 May 2021). The kit components and the protocol of analysis are basically described below:

In total, 100 µL of 1 mg/mL solution of the recombinant RBD protein (a fragment Arg319-Phe541 of SARS-CoV-2 Spike surface glycoprotein, GenBank: QHD43416.1, produced by Hytest LLC, Moscow, Russia) in 100 mM carbonate–bicarbonate buffer (pH 9.6) was added into the microplate wells (Corning, Glendale, CA, USA, cat. #2592) and incubated for 48 h at +22 ± 2 °C. The solution was then removed from the wells, and the plates were washed once with distilled water. Next, 150 μL of blocking solution (0.02 M phosphate-buffered solution containing 5% sucrose, 0.09% sodium caseinate, and 0.05% Twin 20; all the reagents from Merck Millipore, Darmstadt, Germany) was added to the wells and incubated for two hours. After removal of the blocking solution, plates were dried in clean laminar-flow air for 2 h, sealed in vacuum bags, and stored at +4–8 °C until used.

For the determination of the IgG avidity index (AI), test serums, as well as control samples (two positive controls with “high avidity”—AI 62–80% and “low avidity”—AI 23–25% and two negative controls) were incubated in the plate wells in a final dilution of 1/100 in 0.02 M phosphate-buffered solution (pH 7.2) containing 0.2% bovine serum albumin and 0.05% Twin 20. Each sample was incubated in at least two wells (see below).

After incubation for 30 min (+37 ± 2 °C) and washing with an automatic washer (WellWash, Thermo Fisher Scientific, Helsinki, Finland), the pair wells for the same sample were treated with different solutions. The “intact” well was filled with 150 μL of phosphate-buffered saline, while into the “denaturation” well, 150 μL of 4 M urea were added. The 4 M urea concentration was chosen based on both data from the literature [22] and our own verification experiments. The plate was incubated for 10 min at +18–25 °C and then washed.

Next, 100 µL of monoclonal antibodies to human IgG (Sorbent LLC, Moscow, Russia) conjugated with horseradish peroxidase (HRP) in the dilution 1:40,000 was added in the wells and incubated for 30 min at +37°C. After washing, 100 μL of 33 mM citrate buffer solution (pH 4.0) containing 0.01% hydrogen peroxide and 0.5 mM 3,3′,5,5′-tetramethylbenzidine was added. After 15 min, the reaction was stopped by adding 100 μL of 0.5 M sulfuric acid. The optical density (OD) was measured in two-wavelength mode at 450/680 680 nm (Multiskan FC plate photometer, Thermo Fisher Scientific, Helsinki, Finland).

The avidity index (AI) was calculated according to the formula:AI = (OD in the “denaturation” well/OD in the “intact” well) × 100%.

The sample was considered to contain IgG of “low-avidity” at AI ≤ 40%, “high-avidity” at AI ≥ 50%; “gray zone” at AI 40–50% (see the Appendix A in which these cut-offs and their discrimination are described).

In this study, all samples were tested in two repeats, and the mean value for AI was calculated to reduce the variation (for the “SARS-CoV-2-ELISA-IgG plus” kit, the maximum coefficient of variation is declared as CV ≤ 15%).

*Statistics.* The nonparametric Mann–Whitney criterion for median values was used to estimate the occurrence of a parameter in the group (CI 95%) and to compare the groups to each other. Calculations and charts were made with Prism software (GraphPad Software, San Diego, CA, USA). The binomial distribution was used to estimate confidence intervals (CIs) for the proportion of qualitative characteristics in the groups. In all cases, differences with a significance criterion of *p* < 0.05 were considered significant (for details, see captions to the figures in this article).

## 3. Results

### 3.1. Study Groups

Two sets of serum samples positive for IgG to the RBD were included in the study, collected from two groups of patients (Table 1, Appendix A):

Seventy-five samples from clinical patients presenting at state medical institutions in Moscow (Russia) with a mild course of COVID-19. “Mild course” here means a condition that did not require hospitalization; patients were observed as outpatients during the whole period from diagnosis to recovery. Of these, 39 were primarily infected (that is, they had not previously been diagnosed with COVID-19) and 36 were secondarily infected (had a history of the same diagnosis). The primary or secondary infections (past diagnosis) were established using anonymized medical history contained in the unified medical information system of Moscow city (EMIAS [37]);

Ninety-two samples from clinical patients with a severe course of COVID-19 required hospitalization in Moscow state hospitals. Of these, 82 patients were infected primarily and 10 were reinfected.

For patients in both groups, samples were collected after the confirmation of COVID-19 diagnosis (by other medical and laboratory indications). In other words, all the patients were at the early, acute phase of the disease. For patients with reinfection, the time that had passed since the primary diagnosis was known (Table 1). None of the patients in either group had received a SARS-CoV-2 vaccine prior to their first or second infection (data from EMIAS). All samples were collected during June and July of 2021, when the Delta variant of SARS-CoV-2 (B.1.617.2) was totally dominant in Moscow [38]. For some patients, infection by the Delta variant was confirmed by sequencing; see the Appendix A. 

### 3.2. Correlation between the IgG Avidity Index and the Severity of Reinfection

The median avidity index among the subgroup of outpatients who had mild primary COVID-19 and avoided hospitalization was 41.0%, and among mild outpatients who were reinfected, it was 81.4% (Table 1, Figure 1a, significant difference, *p* = 0.0007). This result might have been expected in the reinfected patients, because 213 days (median, Table 1) had passed since the first infection. Thus, their immune system, most probably, had successfully finished maturation during this period (as it had been more than 4 months), and already started to produce high-avidity IgG in response to the repeat SARS-CoV-2 infection, as occurs for many other viruses [13,22,23,24,25,26].

In contrast, patients in the group with a severe (hospitalized) COVID-19 course showed low avidity of IgG, not only in the primarily infected persons (median value of 25%), but also in the reinfected individuals (28.4%) (groups of primarily and secondarily infected patients did not differ significantly, *p* = 0.72, Table 1, Figure 1c). It can be assumed that these patients did not develop effective humoral immunity after their first immunization, and that B-lymphocyte maturation did not occur completely, as described for other coronaviruses [34,36]. Evidence for this is the persistent low avidity of antibodies (AI 28.4%), despite the fact that 217 days had passed between the first and second infections in “severe” patients, which is normally sufficient for the maturation of antibodies (it takes 3 to 4 months to reach maturation, as is evident from the data on outpatients—see Figure 1a above—and as known from other studies [18,19,20]).

The Appendix A contains all the available raw- and meta-data regarding the samples and patients that were examined. 

Importantly, the level of IgG to the RBD itself did not differ in all four cases (primary and reinfection in the outpatient and hospital patient groups: *p* = 0.79 and 0.2, respectively) and was 8.5–7.9 (in PC units) in the first group and >11.7 in the second (Table 1, Figure 1b,d). Thus, the simple measuring of IgG to the RBD had no predictive power regarding the severity of repeated disease. The avidity index, in turn, served as a tool for predicting disease severity. As shown above, where low-avidity antibodies were found in a secondarily infected patient, this was a negative prognostic factor; the detection of high-avidity antibodies suggested a mild course of the disease, which is the main clinically significant result of our study. Interestingly, even among primarily infected patients, those who had a mild course of disease showed slightly higher IgG avidity than patients who were admitted to hospital (Figure 2a, *p* < 0.025, differences are significant).

What is the prognostic power of the AI marker for predicting the severity of repeated COVID-19? To calculate the concordance of the results of AI determination with the clinical course of the disease (i.e., the probability of true prognosis based on the AI measurement), we compared the AIs in the groups of outpatients and hospital patients with repeated disease (Figure 2b). The difference in median AI values in these subgroups was significant (81.45% versus 28.4%, *p* < 0.0001, Table 1). The test system used to determine the AI allows the following interpretation of the results: ‘low-avidity’ sera have AI ≤ 40%; ‘high-avidity’ sera have AI ≥ 50%; and the interval of AI from 40 to 50% is the ‘gray zone’. Among the 36 outpatients with mild reinfection, 30 had high-avidity IgG (AI ≥ 50%), 2 had low-avidity antibodies (AI ≤ 40%), and 4 fell into the ‘gray zone’. Similarly, of the 10 hospital patients with severe reinfection, only 1 had high-avidity antibodies, 8 had low-avidity IgG, and 1 fell into the ‘gray zone’. By applying Bayes theorem to these results (excluding samples with uncertain results), we obtained the following values of the predictive probability of the AI index:-If low-avidity IgG (AI ≤ 40%) is found in a repeat COVID-19 patient, there is an 89 ± 20.5% chance (*p* < 0.05) that the disease will be severe, and the patient will require hospitalization;-If high-avidity IgG (AI ≥ 50%) is found in a repeat COVID-19 patient, there is a 94 ± 7.9% chance (*p* < 0.05) that the disease will be mild, and the patient will not require hospitalization.

## 4. Discussion

In the present study, we showed that low or high avidity of the IgG to SARS-CoV-2 RBD in re-infected patients with COVID-19 correlates with the severity of this repeated disease and may be used as a predictive marker for whether or not the patient will require hospitalization. The avidity here is considered an integral indicator of immune status that the patient had before the infection, which can be investigated retrospectively on the background of a new infection. We recognize that our study had a number of limitations: firstly, the studied groups were relatively small in size (particularly the group of reinfected patients with a severe course of COVID-19, which only included 10 persons); however, that was sufficient to disclose statistically significant differences (see Figure 1 and Figure 2). Next, no vaccinated patients were studied (the reason for this is that in July 2021, the number of previously vaccinated patients with symptoms of COVID-19 in Russia was negligible). Finally, the results and recommendations based on the study of patients with the Delta variant of SARS-CoV-2 (which was prevalent at the time in Moscow [38]) poorly fits for the prognosis of patients with the Omicron variant. Nevertheless, we suggest that the main results of this study will be transferable to new Delta-like variants that are likely to arise in the future.

As noted above, the production of high-avidity IgG is associated with the following events:-A history of successful immunization, in a period of time that allows the immune system to finish the maturation of antibody-producing B-lymphocytes (3–4 months) [18,19,20];-The successful completion of B-lymphocyte maturation (as complete as it can be for beta-coronaviruses at all [34,36]), development of a stable immune response, and formation of a population of memory B-cells [10] which can produce IgG of already high avidity upon new infection;-The improvement of the neutralizing (protective) ability of IgG—because only high-affinity and, therefore, high-avidity antibodies are relevant for virus neutralization, since they can effectively compete with ACE2 for binding to the RBD [27,28,32,34].

This last event is obviously crucial in the severity of repeated disease [21,29,34] since it is directly related to the ability of the existing immunity to prevent virus replication. If the patient’s immune system has failed to complete maturation after the first infection (for reasons that cannot yet be identified, but which may be related to physiological features of the body), the antibodies have lesser protective ability than the antibodies of a patient with completed maturation who has a formed population of memory B-cells. Consequently, the persistence of low-avidity antibodies long after the primary infection is a risk factor for COVID-19 reinfection and its severity that requires the attention of the clinician.

A possible correlation of IgG avidity with the severity of COVID-19 and increased risk of lethal outcomes was shown, in particular, in [39]; however, this had no strong statistical confirmation. Furthermore, [40] demonstrated that the vaccination of recovered patients with low-avidity IgG leads to a statistically significant increase in the avidity of IgG to the S1-antigen of SARS-CoV-2 and thereby decreases the risk of reinfection.

Based on the above, we recommend that the following tests are to be introduced into routine medical practice:(1)If a patient who has had immunization in the past returns with a diagnosis of COVID-19 reinfection, they should be immediately tested for the avidity of their IgG to Spike/RBD. If IgG of low avidity are found (AI ≤ 40%, if the test used is similar to that used in the present study), the patient needs close monitoring since, according to our data, 89 ± 20.5% (*p* < 0.05) of these patients will suffer a severe course of the repeated disease and will require hospitalization.(2)Where possible, all healthy patients immunized against SARS-CoV-2 more than 4 months ago should be tested for their antibody avidity. If the patient still carries IgG of low avidity, preventive revaccination may be recommended [40] because it is highly likely that the patient has not developed sustained immunity and, if reinfected, the patient has a high probability of a severe course of repeated COVID-19.

## Figures and Tables

**Figure 1 viruses-14-00617-f001:**
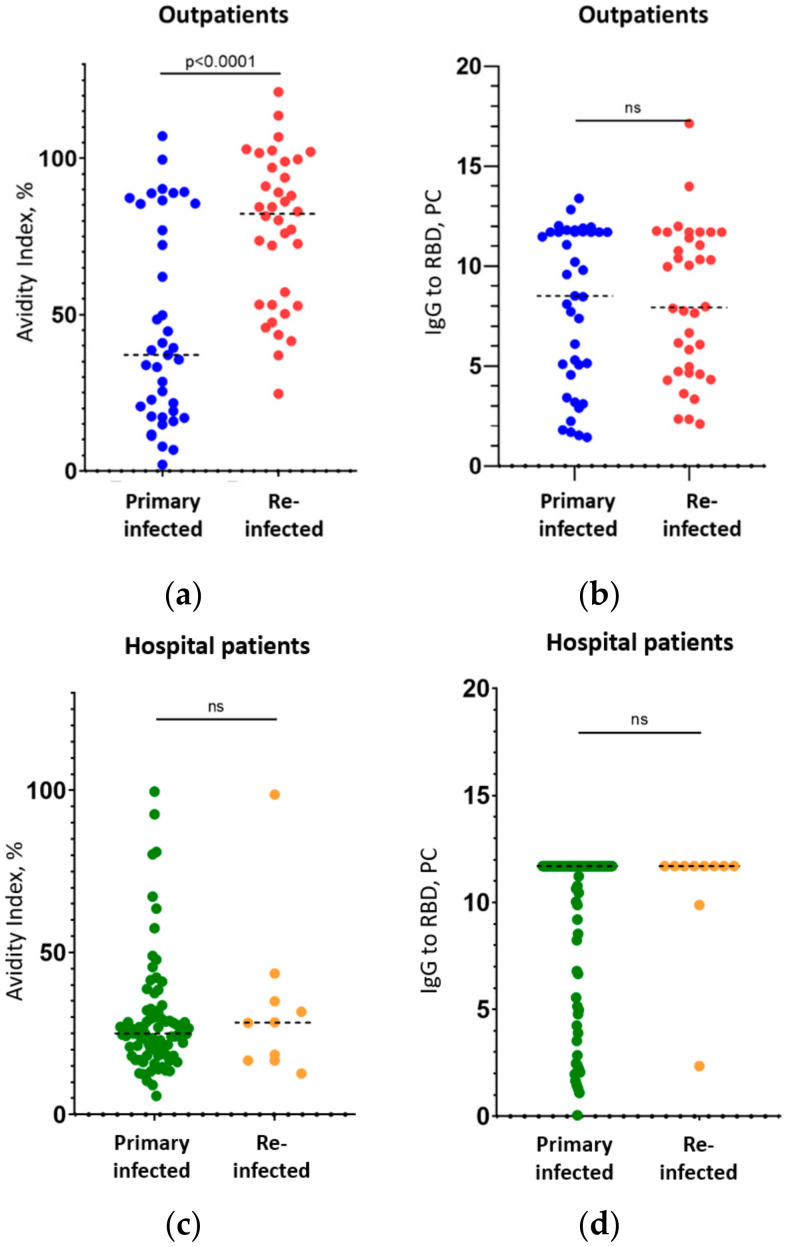
Avidity index (**a**,**c**) and IgG levels (**b**,**d**) in the study groups: outpatients with mild COVID-19 (**a**,**b**) and hospital patients with a severe course of COVID-19 (**c**,**d**). Median values (dashed lines) are shown, as well as a Mann–Whitney test comparison of median values in the groups ”ns”, no significant difference). Numerical values are shown in Table 1.

**Figure 2 viruses-14-00617-f002:**
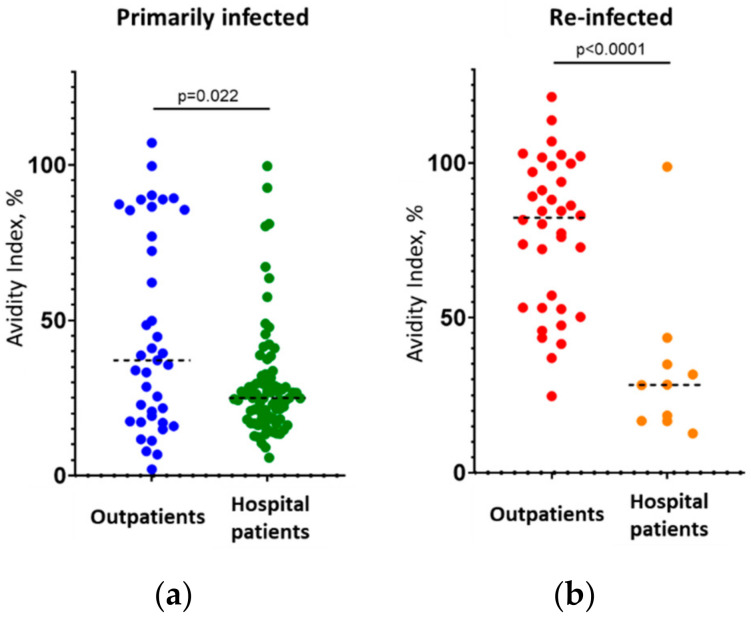
Comparison of AI values in primarily infected (**a**) and reinfected (**b**) COVID-19 patients with mild and severe disease. Notation is the same as that in Figure 1.

**Table 1 viruses-14-00617-t001:** Data for the studied groups of outpatients (with mild course of COVID-19) and hospitalized patients (with severe course) who were primarily or secondarily infected with COVID-19. The confidence intervals (CIs) are shown in brackets.

Group	Outpatients (Mild Course)		Hospital (Severe Course)	
	Primarily Infected	Reinfected	Primarily Infected	Reinfected
Number of patients	39	36	82	10
Median time passed between the first and second COVID-19 infections, days (CI 95%)	-	213 (192–229)	-	217 (191–386)
Median avidity index, % (CI 95%)	37.14 (21.72–62.13)	82.29 (72.1–91.1)	24.99 (22.89–27.05)	28.35 (16.63–43.52)
Median quantity of IgG to the RBD, PC (CI 95%)	8.51 (5.3–11.7)	7.93 (5.82–10.77)	>11.7 (10.45–11.7)	>11.7 (9.88–11.7)

## Data Availability

The data presented in this study are available in Appendix A. Any additional data are available on request from the corresponding author: manuilov@medipaltech.ru.

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
