# Peer review of "Avidity of IgG to SARS-CoV-2 RBD as a Prognostic Factor for the Severity of COVID-19 Reinfection"

_viruses, 2022, doi:10.3390/v14030617_

Round 1

Reviewer 1 Report

The article is interesting, but more studies are needed as well as comparison with the neutralization test that is used for SARS-CoV-2. They want to put the avidity test (antibody functionality) to determine if the patient will be admitted or not. Clinical evaluation table is missing and many more related things... They use ELISA and avidity kits from Russia.... however, that work is in Russian.... I believe this work fits the profile of " major review “. they say that we used the video analysis test and statistical analysis was not performed...... and therefore the need for a comparison study.

Line 2 - Ii is a very interesting point to study .It is important to compare with Neutralization assay .....

Line 19 - disease  without   vacination . Please expalin better ....

Line 24 - It is important to related the population study  with the virus -neutralization effectiveness of the serum used in the present work.

Line 113 - It will be interesting to described about the protol used in this kit . It is  ELISA? 

Line 118 - the analyses of the results are related with the  protocol that are described in literatura as......

Line 129 - It is important to describe details about the  the determination of "low avidity", undetermined results and "high avidity?

Line 164 - Please use the same nomenclature in the table.

Line 164 - What means CIIs?  It is necessary to revise the table

Line 196 -  It will be interesting to show a table or a grafic compare the ELISA title of the patients X ELISA  that analysed at the same time.

Line 222 -  The same  that  I described before  ELISA X Avidity!

Line 224 - Other important point is to use a comparative assay with the patients used as NEUTRALIZATION ASSAY.....

Line 244 - It is necessary to do  Neutralization assay with the sera used to compare with AVIDITY ... the same sample.

Line 253 - It is because is a possible ...It is necessary to do do a NEUTRALIZATION ASSAY .... it is not because statistic . The paper related with assays described in the literature . As a example please show the paper described in experimental level   that  described a  relation with AVIDITY X NEUTRALIZATION Gaspar EB, De Gaspari E. Avidity assay to test functionality of anti-SARS-Cov-2 antibodies. Vaccine. 2021 Mar 5;39(10):1473-1475. doi: 10.1016/j.vaccine.2021.02.003. Epub 2021 Feb 3. PMID: 33581919; PMCID: PMC7857056

Line 260 - Again !It is necessary to compare the sera

used with Neutralization assay for SARS-CoV-2.

Reviewer 2 Report

Manuylov et al. describe the IgG avidity index (AI) as a promising prognostic marker for the disease progression course in case of reinfection with SARS-CoV-2. The authors show a correlation between the AI and the degree of severity of repeated disease and suggest the introduction of screening for AI in routine medical practice.
The manuscript is well written: The introduction gives a good overview, material and methods including statistical analysis applied are described appropriately. Results are well described and presented as figures with clear layout. The discussion section refers to the results found and comparisons to current literature.

  • I would recommend adding a paragraph on vaccination and its putative impact on the AI of SARS-CoV-2 antibodies to the discussion section. Since only unvaccinated individuals were included, it would be interesting to show (or at least discuss) how vaccination affects antibody production and whether there is a difference in avidity between infection acquired or vaccination acquired antibodies.
  • Limitations of the studies should be mentioned including the rather small sample size of study participants.
  • Page 7, line 270: Citation [Struck et al., 2021a] within the text – citation should be given a number (as done for other references) or rewrite the sentence, e.g.: “(…) preventive revaccination as recommended by Struck et al. (…)”.

Reviewer 3 Report

This is a well written paper, although it is rather verbose and pedantic. The experiments are straightforward. The data are limited. I have concerns about experimental design, interpretation, and manuscript presentation.

Experimental design:

  1. In lines 121-122 it states that only a single well was used for each sample. Most published manuscripts contain replicates. For example, I routinely use triplicates, others as many as 6 wells per sample. If these data are to be published we need to see some validation of the variability in this assay (either experimentally by repeat testing of a panel of your samples and showing variance between the two sets of assays, or by literature citation).
  2. Tell us the dilution of the sera used in the ELISA.
  3. Since we are only shown data that has already undergone some degree of manipulation (ie we are not shown or told the ELISA ODs, rather ratios of two parameters), we need to know exactly how the calculations were made. What was background OD vs OD of a strong to moderate signal (ie signal to noise ratio), was background subtracted prior to calculations. Were backgrounds different in urea vs no urea wells?
  4. Were calibration standards, known positive and negative controls run on each assay?

Interpretation:

The authors discuss the avidity assay in terms of prognostic value at the time of reinfection. We are not really told when these samples are taken in terms of onset of disease, ie for how many days did patients have symptoms or have a known exposure before they were tested. As the authors are well aware, timing is everything in measuring a secondary immune response. Yet we are not told that critical piece of information, particularly whether it is the same for outpatient versus hospital patients. Thus to call it of prognostic value is a leap of faith. It might just as well indicate events that have already occurred and could be determined by clinical history. This does not negate the value of your interesting observation, only how it might be interpreted. If you have the data regarding onset of symptoms, it would be an interesting addition.

Editorial comments:

  1. The Introduction is overly long and unnecessary and could be cut severely. Lines 72-75 are a good example of the hyperbole. Most of the paragraph that follows is so basic that anyone reading the paper will know it already. We know about avidity and how it works and measured. Similar discussions could be trimmed in results and discussion as well.
  2. Lines 95-98, assumes that the only mechanism of neutralization is Ab competition for ACE2. Other mechanisms of neutralization by anti-RBD antibodies have been demonstrated (eg. altering RBD orientation).
  3. Figure 2 seems to be exactly the same data shown in figure 1, but displayed differently. The problem is the labels. The red dots are labeled “reinfected outpatients” in figure 1, and “primary infected” in figure 2. I think the labels in figure 2 should be “outpatients” and “hospitalized”. But in either case figure 2 is unnecessary since it is a repetition of data. You can just state in the text that the difference between the groups is significant at p<0.0001.

Round 2

Reviewer 1 Report

I agree with the changes to the manuscript and I believe that the questions were answered.

Reviewer 3 Report

The authors have responded well to a series of questions I have raised.